

# Towards FAIR protocols and workflows: the OpenPREDICT use case

Remzi Celebi[1,*], Joao Rebelo Moreira[2,*], Ahmed A. Hassan[3], Sandeep Ayyar[4], Lars Ridder[5], Tobias Kuhn[2] and Michel Dumontier[1]

[1] Institute of Data Science, Maastricht University, Maastricht, Netherlands
[2] Computer Science, VU University Amsterdam, Amsterdam, Netherlands
[3] Pharmacology & Personalised Medicine, Maastricht University, Maastricht, Netherlands
[4] Medical Informatics, Stanford University, Palo Alto, CA, United States of America
[5] Netherlands eScience Center, Amsterdam, Netherlands
[*] These authors contributed equally to this work.

Corresponding authors
Remzi Celebi,
remzi.celebi@maastrichtuniversity.nl
Joao Rebelo Moreira,
j.luizrebelomoreira@utwente.nl

## ABSTRACT

It is essential for the advancement of science that researchers share, reuse and reproduce each other's workflows and protocols. The FAIR principles are a set of guidelines that aim to maximize the value and usefulness of research data, and emphasize the importance of making digital objects findable and reusable by others. The question of how to apply these principles not just to data but also to the workflows and protocols that consume and produce them is still under debate and poses a number of challenges. In this paper we describe a two-fold approach of simultaneously applying the FAIR principles to scientific workflows as well as the involved data. We apply and evaluate our approach on the case of the PREDICT workflow, a highly cited drug repurposing workflow. This includes FAIRification of the involved datasets, as well as applying semantic technologies to represent and store data about the detailed versions of the general protocol, of the concrete workflow instructions, and of their execution traces. We propose a semantic model to address these specific requirements and was evaluated by answering competency questions. This semantic model consists of classes and relations from a number of existing ontologies, including Workflow4ever, PROV, EDAM, and BPMN. This allowed us then to formulate and answer new kinds of competency questions. Our evaluation shows the high degree to which our FAIRified OpenPREDICT workflow now adheres to the FAIR principles and the practicality and usefulness of being able to answer our new competency questions.

## INTRODUCTION

Reproducible results are one of the main goals of science. A recent survey, however, showed that more than 70% of researchers have been unsuccessful in reproducing another research experiment and more than 50% failed to reproduce their own research studies (*Baker, 2016*).

The rate of non-reproducibility for pharmacological studies is particularly worrying. Together with their high costs and their high rate of failure (around 90%), this highlights

the need for new approaches in drug discovery (*Scannell et al., 2012*). For these reasons, we chose pharmacology as the field to apply and test the approach we will introduce below. Specifically, we will be looking into drug repositioning, where small molecules approved for one indication are repurposed for a new indication. Drug repositioning is gaining recognition as a safe, effective and lower-cost approach to uncover new drug uses (*Ashburn & Thor, 2004*; *Sleigh & Barton, 2010*). The availability of public data, both in the form of literature curated knowledge and omics data has created exciting opportunities for computational drug repositioning. For instance, gene expression data in repositories such as the Gene Expression Omnibus (GEO) enable the analysis of correlations between drug and gene expression - termed the Connectivity Map approach - to find chemicals that may counter cellular disorders (*Barrett & Edgar, 2006*), including Alzheimer's, and small cell lung cancer (*Lamb et al., 2006*; *Sirota et al., 2011*). More sophisticated approaches use network analysis and machine learning to efficiently combine drug and disease data (*Cheng et al., 2012*; *Gottlieb et al., 2011*; *Hoehndorf et al., 2013*; *Wu et al., 2013*; *Bisgin et al., 2014*).

The ability to reproduce original research results is contingent on the availability of the original data, methods and results. The FAIR principles (*Wilkinson et al., 2016*), describe a set of requirements for data management and stewardship to make research data Findable, Accessible, Interoperable, and Reusable. Ongoing efforts on FAIR cover data policies, data management plans, identifier mechanisms, standards and data repositories (*Collins et al., 2018*). Highly diverse communities, from the biomedical sciences to the social sciences and humanities, are now working towards defining standards for publication and sharing of data. In anticipation, new methods and infrastructure are needed to facilitate the generation of FAIR data and workflows.

Here, we describe a methodology to publish scientific workflows as FAIR data. We are using the term workflow here to include computational steps implemented in software but also manual steps, such as manual data cleaning steps or wet-lab activities. We evaluate our method by applying it to the PREDICT drug repositioning workflow. Based on this example, we will try to answer our research question of how we can use existing vocabularies and techniques to make scientific workflows more open and FAIR, with a particular focus on the interoperability aspect. The main contributions of this paper are (a) general guidelines to make scientific workflows open and FAIR, focusing on the interoperability aspect, (b) the OpenPREDICT use case, demonstrating the open and FAIR version of the PREDICT workflow, (c) new competency questions for previously unaddressed reproducibility requirements, and (d) evaluation results on the practicality and usefulness of our approach.

## BACKGROUND

Below, we refer to the most relevant background with respect to reproducibility, workflow systems, and applying FAIR to workflows.

### Scientific Workflows and Reproducibility

According to the Descriptive Ontology for Linguistic and Cognitive Engineering (DOLCE) (*Borgo & Masolo, 2010*), a workflow is a "plan that defines role(s), task(s), and a

specific structure for tasks to be executed, usually supporting the work of an organization",
and a plan is a description of instructions with an explicit goal. A scientific workflow,
therefore, is such a plan that implements scientific methods to work towards the general
goal of scientific knowledge gathering and organization. Certain scientific workflows can
be automated through workflow systems, which are software systems that enable the
representation and execution of structured tasks.

The lack of relevant details in the published descriptions of scientific workflows
(*Vasilevsky et al., 2013*) is an important factor contributing to the non-reproducibility
rates of 64% in pharmacology (*Ioannidis, 2005a*; *Prinz, Schlange & Asadullah, 2011*), 89%
in cancer research (*Begley & Ellis, 2012*), and 66% in psychology (*Klein et al., 2014*). A
recent analysis of over 1.4 million Jupyter notebooks (available in GitHub) found that
only 24.11% of the notebooks could be executed without errors and only 4.03% produced
the same results (*Pimentel et al., 2019*). As a consequence, it has been reported that data
scientists spend 19% of their time finding, understanding and accessing datasets, and 60%
of their time cleaning and organizing these datasets to use in their studies (*CrowdFlower,
2016*). Thus, only 20% of the time is left for data scientists to spend on their core activities,
such as mining data, refining algorithms, building training sets and analyzing the results.

## Workflow systems

To tackle the workflow decay phenomenon (*Hettne et al., 2012*), a number of recent
initiatives are targeting the improvement of the reproducibility of computational workflows
for example the Common Workflow Language (CWL) (https://www.commonwl.org/) and
the Workflow Description Language (WDL) (https://openwdl.org/), which have become
the de facto standard for syntactic interoperability of workflow management systems.
CWL and WDL are aimed to exchange and run computational workflows reproducibly
in different environments. They are designed to separate the workflow description from
its execution. In order to improve semantic interoperability and connect workflows
to real-world entities in a systematic way, additions of semantic models and methods
have been proposed, for example the Workflow4ever project with its Research Objects
method (*Belhajjame et al., 2015*).

Provenance is an important aspect of workflows, which can be classified into prospective
provenance, retrospective provenance, and workflow evolution provenance. Prospective
provenance refers to the specifications or "recipes" that describe the workflow steps and
their execution order, typically as an abstract representation of these steps (protocols),
as well as expected inputs and outputs (*Cohen-Boulakia et al., 2017*). Retrospective
provenance refers to the information about actual workflow executions that happened
in the past, including the concrete activities that consumed inputs and produced outputs,
as well as information about the execution environment (*Khan et al., 2019*). Workflow
evolution provenance refers to tracking the versions of workflow specifications and the
respective data, as the workflow specification is changed and improved over time.

A number of models and methods have been developed to capture these different kinds
of provenance. The PROV ontology (*Lebo et al., 2013*) provides the vocabulary and model
for provenance in general, which can be used in conjunction with top-level ontologies such

as DOLCE (*Borgo & Masolo, 2010*) and other general vocabularies such as Dublin Core and schema.org. Several approaches have been proposed to apply PROV to workflows, such as the Open Provenance Model for Workflows (OPMW) (*Moreau et al., 2008*), P-PLAN (*Garijo & Gil, 2012*), and CWLProv (*Soiland-Reyes et al., 2018*). Other notable examples include ProvBook and the Reproduce-me ontology (*Samuel & König-Ries, 2018a*; *Samuel & König-Ries, 2018b*) for workflows in Jupyter notebooks, the ML-Schema ontology for machine learning workflows (*Correa Publio et al., 2018*), the Publishing Workflow Ontology (PWO) for workflows in scientific publications (*Hartanto, Sarno & Ariyani, 2017*), and the Business Process Modelling Notation (BPMN) to specify business processes (*Rospocher, Ghidini & Serafini, 2014*). Other approaches, such as SMART protocols (*Giraldo et al., 2017*) and protocols.io, target the description of laboratory protocols.

### Applying FAIR to workflows

The FAIR principles have received significant attention, but we currently lack overarching approaches to align them with scientific protocols and workflows in a broad sense. Making a workflow FAIR-compliant entails that general-purpose software can interpret it and understand its context. The application of FAIR in healthcare, for example, has shown that these principles boost data-driven applications that require the integration of data coming from different sources, achieving ''interoperability without the need to all speak exactly the same language'' (*Imming et al., 2018*). Recent initiatives have outlined how FAIR can be applied to software (*Neil & Daniel, 2018*; *Lamprecht et al., 2019*), contributing towards the goal of applying FAIR not just to input and output data, but to the entire process in between, in order to solve the current problem that even human experts are often unable to reconstruct the specific steps and parameters of a workflow from what is published in scientific articles (*Vasilevsky et al., 2013*).

The FAIRification *Jacobsen et al. (2019)* consists of a number of steps required to transform an existing data element to its FAIR version, typically leveraging the RDF technology for the interoperability aspect. RDF is a broadly applicable formal language to achieve the semantic interoperability principle I1. FAIRification starts by retrieving the non-FAIR data from its sources. Subsequently these datasets are analyzed to understand the data structures and how they are mapped to concepts from the domain. The next step, semantic modelling, is a major activity comprising semantic harmonisation and integration, requiring the reuse and/or creation of models compliant with the FAIR principles. Once the dataset is aligned with semantic definitions, it can be expressed in RDF and augmented with metadata. The last step is to store the FAIRified data into a findable and accessible manner.

## THE FAIR WORKFLOWS APPROACH

In this section we describe our workflow representation requirements, with a special focus on the coverage of manual steps, different workflow abstraction levels, and versioning on all these levels. We formulate these requirements as competency questions and present a

configuration of elements from existing semantic models as a unified model to answer these competency questions.

## Requirements and Competency Questions

With the help of structured interviews with data scientists and a gap analysis of the literature, we formulated user requirements for the reproducibility of workflows (the details of the interviews are given in Appendix S1).

The interviewees stated that they experience many challenges in reproducing their or others' work, due to the lack of details of workflow steps, data cleaning and filtering. Also essential information, such as processing parameters or design details needed to reproduce the results, is often missing. Some of these requirements are already covered by existing approaches while others have not been addressed so far. The interviews indicated that the definitions of manual processes of workflows are usually missing or incomplete, which is a requirement poorly addressed by computational workflow approaches. Often, software libraries, packages and versions of tools used are not explicitly recorded. The interviewees suggested making metadata of the datasets accessible, add richer prospective and retrospective provenance and allowing for fine-grained workflow versioning linked to outputs produced during distinct executions. A unanimous recommendation was to allow for the separate input of relevant workflow parameters, so that one can run the same workflow multiple times with different processing options without having to change the workflow itself.

The representation of software environment details (e.g., the used libraries and packages) is already addressed by some of the surveyed semantic models, like Workflow4ever, CWLProv and Reproduce-me. We checked the capabilities of the existing semantic approaches to address the needs collected from the interviews. We concluded that none of the related work could completely address all the requirements together. The missing parts can be put in three main categories: (CQ1) Manual steps description and executions; (CQ2) abstraction levels of workflows; and (CQ3) versioning of executed workflows. Therefore, we propose the following additional sets of competency questions (CQ) to cover these missing parts:

The first group of questions (CQ1) is about manual steps:

CQ1.1   Which steps are meant to be executed manually and which to be executed computationally?

CQ1.2   For the manual steps, who are the agents responsible to execute them (individuals and roles)?

CQ1.3   Which datasets were manually handled and their respective formats?

CQ1.4   What are the types of manual steps involved?

The second group (CQ2) is about instantiation of general workflows by more specific ones:

CQ2.1   What are the main steps of a general workflow?

CQ2.2   What are the steps of a specific workflow and how are they described?

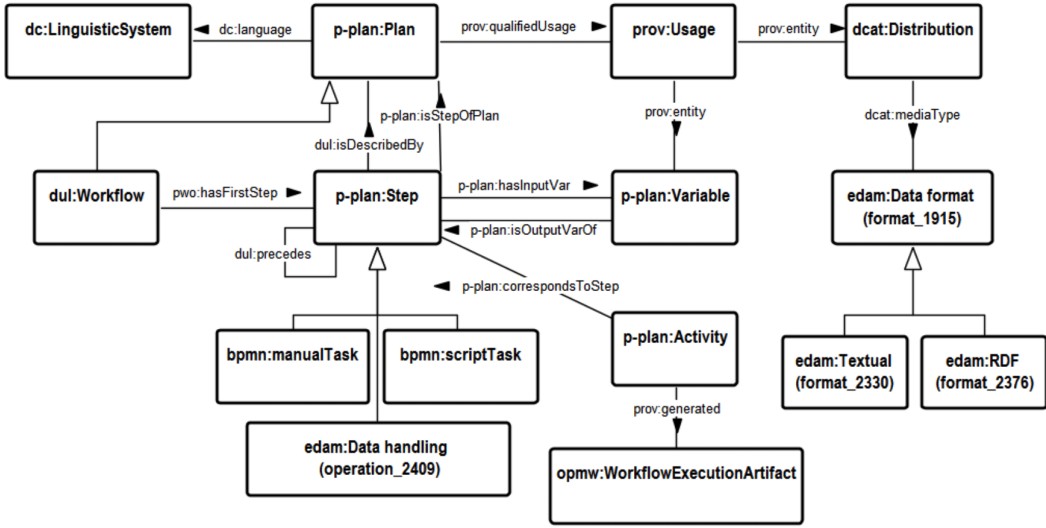

**Figure 1** Unified semantic model for workflows.

CQ2.3 What higher-level description instantiated a certain workflow step?
CQ2.4 Who or what method made the instantiation of a semantic/meta level description of a step into an executable workflow step?

The third group (CQ3) are questions about versioning of workflows and their executions:

CQ3.1 What are the existing versions of a workflow and what are their provenances?
CQ3.2 Which instructions were removed/changed/added from one version to another?
CQ3.3 Which steps were automatized from one version to another?
CQ3.4 Which datasets were removed/changed/added for the different versions?
CQ3.5 Which workflow version was used in each execution and what was generated?

To the best of our knowledge, none of the previous research on semantic modelling of workflows (or protocols/processes) addresses all these requirements together. In few cases some semantic models only partially cover some questions, as explained in the prior section.

## Unified model

From the study of the diverse existing semantic models for workflows and protocols, we compiled a unified conceptual model covering the elements required to answer our competency questions. For this, we applied the ontology-driven conceptual modelling approach (*Guizzardi et al., 2015*), which is based on the Unified Foundational Ontology (UFO) and its ontological language OntoUML (*Moreira et al., 2016*).

Figure 1 illustrates the main elements of our unified model (https://w3id.org/fair/plex), which is primarily based on DOLCE Ultra Lite (DUL), PROV, P-PLAN and BPMN 2.0.

The most relevant ontology used is P-PLAN, which provides an abstract terminology of the main building blocks to describe plans.

The *p-plan:Plan* category is the core element of our unified model and is the class used to classify any type of instruction. It allows for the composition of instruction by means of smaller steps (*p-plan:Step*) that have input and output variables (*p-plan:Variable*). With the *pwo:hasFirstStep* property, we can indicate the first step of a plan, and with *dul:precedes*, we can indicate whenever a step precedes another, thereby enabling the representation of sequential and parallel steps.

We decouple a particular step within a workflow from its instruction with the pattern *p-plan:Step dul:isDescribedBy p-plan:Plan*, where each step always points to one plan. This approach allows us to separate the workflow steps, enabling the reuse of instructions by different workflows. Therefore, in our approach a step is a lightweight object (like a pointer) that serves only for ordering of instructions without coupling them to the specific workflow. Besides that, we use the *dul:isDescribedBy* property as a self-relationship of *p-plan:Plan*, to represent that an instruction describes another instruction in a different abstraction level. With this approach, we can represent anything from high-level abstract protocols to concrete and executable workflow steps, and the links between these levels. This can be used to first represent the general protocol and then move to the definition of the executable steps akin to the common software engineering phases of specification and implementation. Our model can however also be used in the other direction to extract a new common protocol from similar existing concrete workflows. At the more abstract levels, instructions are written in a natural language like English (or possibly pseudo-code), whereas at the lowest level, we find the executable specifications, which can be written in a programming language and thereby automatically executable. Alternatively, at the lowest level instructions can be in natural language, such as for wet-lab instructions, which can naturally only be executed in a manual fashion. For example, the first general step of a specification of a machine learning pipeline like OpenPREDICT (to which we will come back to shortly) might be to "load all features and gold standard" (a *p-plan:Plan*). The concrete execution of this general step is described by four concrete and executable steps (written in a language such as Python), each having a link (*dul:isDescribedBy*) to the general description of the step.

We use the BPMN 2.0 ontology for the representation of manual and computational activities with *bpmn:ManualTask* and *bpmn:ScriptTask*, which we both define as subclasses of *p-plan:Step*. With this approach, the modeller can therefore include manual and automated steps in the same workflow. More specific classes can be used for particular workflow systems, such as *reprod:Cell* as a kind of *bpmn:ScriptTask* describing a code cell in a Jupyter Notebook.

We follow the FAIR Data Point specification (https://github.com/FAIRDataTeam/FAIRDataPoint-Spec) for the representation of datasets (input and output) through the *dcat:Dataset* element, which should be linked to the available distributions (*dcat:Distribution*) through the *dcat:distribution* property, and the URL to download the distribution is represented with *dcat:downloadURL*. We improved this approach with data formats from the EDAM ontology through the *dcat:mediaType* property. We use

*prov:qualifiedUsage* for variable bindings. For example, the instruction (*p-plan:Plan*) to "download a dataset and save it in the local environment" has a link (*prov:qualifiedUsage*) to the "binding the online dataset to a local variable" (*prov:Usage*), which represents the connection between the dataset distribution (*dcat:Distribution*) and the local variable (*p-plan:Variable*) through instances of the *prov:entity* properties. For the representation of retrospective provenance, i.e., information about prior executions of a workflow, we follow the P-PLAN approach by using *p-plan:Activity* and linking it to the steps with *p-plan:correspondsToStep*.

To represent the roles of the different involved agents (such as people and software), we use the agent associations as defined in PROV. For example, the Jupyter Notebook (*prov:SoftwareAgent*) was used as execution environment (*prov:Role*) for all computational steps of the OpenPREDICT workflow.

Furthermore, as a practical design decision, we extended the notion of *prov:Association* for endurants, so the modeller can apply the association pattern similarly to the perdurant way, i.e., use the property *prov:hadPlan* from *p-plan:Association* to *p-plan:Plan* instead of the relation from *prov:Activity* through *prov:qualifiedAssociation*. Therefore, this approach allows the modeller to represent the association of agent roles to an instruction. For example, Remzi is the OpenPREDICT main developer, so the "Remzi as developer of OpenPREDICT" (*prov:Association*) links to (a) the "Developer" (*prov:Role*) through *prov:hadRole* property, (b) the Remzi object (a *prov:Agent*) through *prov:agent*; and (c) all OpePREDICT instructions (*p-plan:Plan*), through *prov:hadPlan*. Notice that, although the terminology of these properties targeted the perdurant aspect (*prov:Activity*), these properties are also useful for the endurant aspect. Ideally, they should have the adequate endurant terminology, so instead of prov:hadPlan, it should be "*prov:hasPlan*" (similarly for *prov:hadRole* too).

One of the most important links is the one between a workflow execution and its created outputs. For this, we specialized the PROV approach by using *prov:generated* to link a workflow activity (*p-plan:Activity*) to an output artefacts (*opmw:WorkflowExecutionArtifact*). Therefore, each step execution can generate workflow execution artifacts. To represent the specifics of machine learning workflows, we moreover use the ML-Schema ontology (mls), such as to specify the trained model and its evaluation measures (via *mls:ModelEvaluation* and *mls:EvaluationMeasure*). For example, it can be used to specify the accuracy of the models that were trained during different executions.

For the representation of versioning, finally, we use *dc:hasVersion* to assign version identifiers and *prov:wasRevisionOf* to link to the previous versions, and apply this to all relevant elements, including workflows, instructions, software, and datasets.

## Case study topic

We evaluate our approach below with a case study of a computational drug repurposing method based on machine learning, called PREDICT (*Gottlieb et al., 2011*). PREDICT is one of the most frequently cited drug repurposing methods and provides a ranking of drug-disease associations on the basis of their similarity to a set of known associations.

PREDICT has reported a high AUC (0.90) for predicting drug indications, though neither the original data nor the software to produce the results are available.

The features for the drug prediction classifier included five drug–drug similarity measures and two disease–disease similarity measures. The similarities between drugs were calculated based on molecular fingerprints, common side effects of drugs, target protein sequence alignment, semantic similarity of target genes of drugs in the Gene Ontology, and closeness of target proteins in human protein–protein interaction network. For the disease aspect, two disease–disease similarities were calculated based on medical description of diseases and semantic similarity of disease terms in the Human Phenotype Ontology. The method transforms drug–drug and disease–disease similarities into integrated features to be used for a logistic regression training.

For evaluating the performance of the logistic regression, 10-fold cross-validation was used in two different ways: one in which 10% of drugs are hidden and one in which 10% of associations are hidden. In the first strategy, 10% randomly selected drugs in the gold standard and the known indications associated with them were removed. The positive training set consisted of the remaining 90% of drugs and the indications associated with them. The negative training set consisted of randomly generated drug-disease associations which were not in the positive set. For the second strategy, the known associations were divided into 90% positive training and 10% positive test sets, while negative training and test sets were built using randomly generated drug-disease associations from respective sets.

In the next section, we report on the application of our approach to this use case.

## OPENPREDICT CASE STUDY

As case study, we took the original PREDICT workflow, as introduced above, and transformed it with our approach to make it open and FAIR. We therefore call the resulting workflow OpenPREDICT. It implements the same steps of the original PREDICT, i.e., five drug–drug similarity and two disease–disease similarity measures were used to train a logistic regression classifier to predict potential drug-disease association (see Fig. 2). Therefore, we follow the same general protocol of these four steps:

1. **Data preparation:** In this step, the necessary dataset is collected and preprocessed.
2. **Feature Generation:** In this step, we generate features from the collected data sets. Drug-drug and disease-disease similarity scores were combined by computing the weighted geometric mean. Thus, we combine five drug-drug similarity measures and two disease-disease similarity measures, resulting in 10 features.
3. **Model Training:** In this step, the generated features from the previous step are to be used to train in a simple logistic classifier.
4. **Model Evaluation:** This step uses two different cross-validation approaches: one where 10% of drugs is hidden and one where 10 % of associations is hidden for testing. ROC AUC, AUPR, accuracy, precision and F-score of the classifier on test data are reported.

Below we explain how we made a FAIR version of PREDICT's input data and then show how we used our approach to model the OpenPREDICT workflow that is consuming this

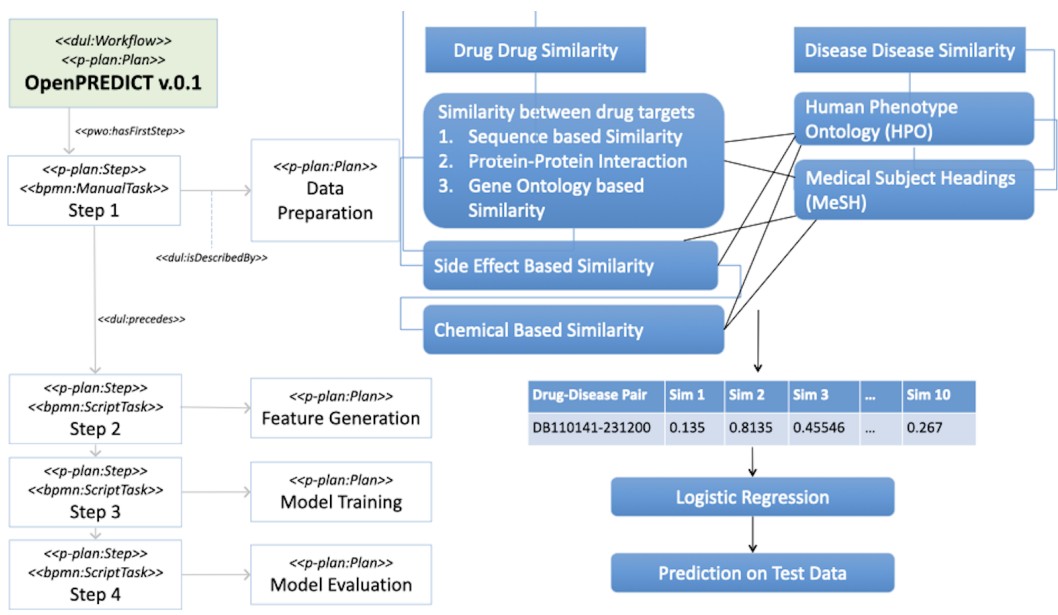

**Figure 2** OpenPREDICT Workflow (version 0.1) with manual and computational steps.

data. The implementation and the workflow description of OpenPREDICT are available on GitHub (https://github.com/fair-workflows/openpredict).

## FAIRified data collection

Since the original data used in PREDICT is not publicly available, we collected data from open sources and made it FAIR with Linked Data (*Bizer, Heath & Berners-Lee, 2009*) representations. We obtained data about drugs, drug targets, drug chemical structure, and drug target sequence from DrugBank (*Wishart et al., 2008*), and additional drug targets from the KEGG dataset (*Kanehisa et al., 2007*). The SIDER dataset (*Kuhn et al., 2010*) was used for drug side effects and the HGNC and the GOA datasets (*Gray et al., 2015*; *Barrell et al., 2009*) were used for gene identifier mapping and gene ontology (GO) annotation respectively. We used Linked Data versions of the above-mentioned datasets from Bio2RDF (*Callahan, Cruz-Toledo & Dumontier, 2013*), which is an influential resource for the biomedical sciences, providing a network of data collected from several major biological databases. On to of that, we used the supplementary file provided by *Menche et al. (2015)* for protein–protein interactions and disease phenotype annotations that link HPO terms to OMIM diseases (https://hpo.jax.org/app/download/annotation). MeSH annotations were collected from (*Caniza, Romero & Paccanaro, 2015*) (https://paccanarolab.org/disease_similarity) and annotations were also obtained by NCBO annotator API (*Noy et al., 2009*) using the OMIM disease description.

The data that was not yet in a Linked Data format were converted to RDF with a FAIRification process (*Jacobsen et al., 2019*). We kept the copies of the retrieved non-RDF datasets in our GitHub repository to prevent the data access issues that may arise if data sources are unavailable. We also stored the collected datasets in a triplestore and created

**Table 1  All datasets used in OpenPREDICT version v0.1 and v0.2.**

| Dataset file | Date retrieved | Data format | Download URL |
|---|---|---|---|
| Bio2RDF r4 datasets (Drugbank, KEGG, HGNC, SIDER and GOA) | 2019-08-15 | .nq (RDF) compressed as .gz | https://download.bio2rdf.org/#/release/4/ |
| PREDICT drug indication gold standard | 2019-08-15 | .tab with tabular separator | https://www.ncbi.nlm.nih.gov/pmc/articles/PMC3159979/bin/msb201126-s4.xls |
| Pubchem-Drugbank mappings | 2019-08-15 | .tab with tabular separator | https://raw.githubusercontent.com/dhimmel/drugbank/gh-pages/data/mapping/pubchem.tsv |
| Protein-protein interactions | 2019-08-15 | .txt with tabular separator | https://science.sciencemag.org/highwire/filestream/628238/field_highwire_adjunct_files/1/Datasets_S1-S4.zip |
| HPO Phenotype annotations | 2019-08-15 | .tab with tabular separator | http://compbio.charite.de/jenkins/job/hpo.annotations/lastSuccessfulBuild/artifact/misc/phenotype_annotation.tab |
| †MESH Phenotype annotations | 2019-08-15 | .tab with tabular separator | http://www.paccanarolab.org/static_content/disease_similarity/mim2mesh.tsv |
| MESH Phenotype annotations (BioPortal) | 2019-08-15 | .txt file | https://raw.githubusercontent.com/fair-workflows/openpredict/master/data/external/meshAnnotationsFromBioPorttalUsingOMIMDesc.txt |

SPARQL queries to access the triplestore in order to produce the features for PREDICT's method.

Our OpenPREDICT workflow has two versions (0.1 and 0.2). In the first, we experimented with the FAIRifier tool with the two inputs that are provided as text files, i.e., (protein–protein interactions in human interactome) and (disease phenotypic descriptions). Besides the formalization of the manual steps through our approach, we also provide guidelines for the manual steps. In the second, we wrote Python scripts for FAIRificiation process of these datasets, evolving most of the manual steps to computational ones. Table 1 summarizes the list of all datasets used in version v0.1 and v0.2. OpenPREDICT v0.2 also uses a different MESH annotation dataset for disease similarity (indicated with † in Table 1)

For this FAIRification process, the data needs to be mapped to formal semantic models. In our case, important concepts included "protein-protein interaction" from Bioportal, "protein interactions" from EDAM (*edam:topic_0128*), the Bio2RDF properties *bio2rdf:interactor_a* and *bio2rdf:interactor_b* for the gene interactors representing the role of a gene in a protein–protein interaction, and "disease" and "has phenotype" from SIO (SIO_010299 and SIO_001279).

## OpenPREDICT workflow representation

Figure 2 illustrates the main steps of the OpenPREDICT workflow, in which the main protocol is represented as a *dul:Workflow* and a *p-plan:Plan*, with version set through the *dc:hasVersion* property. The workflow consists of four steps: data preparation, feature generation, model training and evaluation, and presentation of results. Each one is defined

by (*dul:isDescribedBy*) its own *p-plan:Plan*. In the first version of OpenPREDICT (0.1) all steps within the data preparation were manual (*bpmn:ManualTask*), as the FAIRification process and the preparation steps on data that were already provided as RDF. The second version of OpenPREDICT (0.2) automated most of these manual steps, requiring less human intervention. We will now go through some of the most important aspects of this representation.

### Prospective provenance

We decoupled the workflow steps from the instructions, linking a *p-plan:Step* to a *p-plan:Variable* through *p-plan:hasInputVar* and *p-plan:hasOutputVar*, while the *p-plan:Plan* links to the *prov:Usage* through the *prov:qualifiedUsage* property, describing how to bind the variable to other resources. This is an example:

```
opredict:Step_Download_Drugbank_dataset
  rdf:type bpmn:ManualTask ;
  rdf:type edam:operation_2409 ;
  rdf:type p-plan:Step ;
  p-plan:hasOutputVar opredict:Variable_Drugbank_dataset_online ;
  p-plan:isStepOfPlan opredict:Plan_Main_Protocol_v01 ;
  dul:isDescribedBy opredict:Plan_Download_Drugbank_dataset ;
  dul:precedes opredict:Step_Save_Drugbank_dataset ;
  rdfs:label "Download Drugbank dataset" ;
.

opredict:Plan_Download_Drugbank_dataset
  rdf:type p-plan:Plan ;
  dc:description "Download Drugbank dataset" ;
  dc:language :LinguisticSystem_xsd_language_English ;
  rdfs:label "Download Drugbank dataset" ;
  prov:qualifiedUsage opredict:
      Usage_Fetch_download_Drugbank_dataset_to_variable ;
.

opredict:Usage_Fetch_download_Drugbank_dataset_to_variable
  rdf:type prov:Usage ;
  rdfs:label "Link variable to download Drugbank dataset" ;
  prov:entity opredict:Distribution_release-4-drugbank-drugbank.nq.gz;
  prov:entity opredict:Variable_Drugbank_dataset_online ;
.

opredict:Distribution_release-4-drugbank-drugbank.nq.gz
  rdf:type dcat:Distribution ;
  rdfs:label "release/4/drugbank/drugbank.nq.gz" ;
  dcat:downloadURL "http://download.bio2rdf.org/files/release/4/drugbank/
      drugbank.nq.gz" ;
  dcat:mediaType opredict:DataFormat_nq_compressed_gz ;
.

opredict:Variable_Drugbank_dataset_online
  rdf:type p-plan:Variable ;
  rdfs:label "Drugbank dataset online" ;
.
```

### Retrospective provenance

We represent the concrete executions that happened and the concrete output that was generated with a *p-plan:Activity* that is linked to a *p-plan:Step* through the *p-plan:correspondsToStep* property and to the outputs (*opmw:WorkflowExecutionArtifact*) through *prov:generated*. Each output has a value (e.g., accuracy rate) and is linked to

*prov:Generation* through the *prov:qualifiedGeneration* property, which specifies when the generation occurred with (*prov:atTime*). This is an example:

```
opredict:
    Activity_Model_preparation_train_and_evaluation_Execution_1546302862
  rdf:type p-plan:Activity ;
  p-plan:correspondsToStep opredict:
    Step_Model_preparation_train_and_evaluation ;
  prov:generated opredict:ModelEvaluation_Accuracy_Execution_1546302862 ;
  prov:generated opredict:
    ModelEvaluation_AveragePrecision_Execution_1546302862 ;
  prov:generated opredict:ModelEvaluation_F1_Execution_1546302862 ;
  prov:generated opredict:ModelEvaluation_Precision_1546302862 ;
  prov:generated opredict:ModelEvaluation_Recall_Execution_1546302862 ;
  prov:generated opredict:ModelEvaluation_RocAuc_Execution_1546302862 ;
.

opredict:ModelEvaluation_Accuracy_Execution_1546302862
  rdf:type mls:ModelEvaluation ;
  dc:description "0.833336" ;
  mls:specifiedBy opredict:EvaluationMeasure_PredictiveAccuracy ;
  prov:qualifiedGeneration opredict:Generation_Execution_1546302862 ;
.

opredict:Generation_Execution_1546302862
  rdf:type prov:Generation ;
  prov:atTime "2019-01-01T00:02:31.011"^^xsd:dateTime ;
```

### Versioning of workflows

We track the modification across the two versions with the *dc:hasVersion* property on the level of a *dul:Workflow*, *p-plan:Plan*, *dc:LinguisticSystem*, and *prov:SoftwareAgent*. Furthermore, we use the *prov:wasRevisionOf* property to link to the previous version. This is an example:

```
opredict:Plan_Main_Protocol_v02
  rdf:type p-plan:Plan ;
  rdf:type dul:Workflow ;
  dc:created "2019-05-15" ;
  dc:creator opredict:Agent_Remzi ;
  dc:description "OpenPREDICT Main Protocol v.0.2" ;
  dc:hasVersion "0.2" ;
  dc:language :LinguisticSystem_xsd_language_English ;
  dc:modified "2019-07-03" ;
  pwo:hasFirstStep opredict:Step_Prepare_Input_Data_Files_v02 ;
  rdfs:label "Main Protocol v.0.2" ;
  prov:wasAttributedTo opredict:Agent_Remzi ;
  prov:wasRevisionOf opredict:Plan_Main_Protocol_v01 ;
.

opredict:Plan_Main_Protocol_v01
  rdf:type p-plan:Plan ;
  rdf:type dul:Workflow ;
  dc:created "2018-11-27" ;
  dc:creator opredict:Agent_Remzi ;
  dc:description "OpenPREDICT Main Protocol v.0.1" ;
  dc:hasVersion "0.1" ;
  dc:language :LinguisticSystem_xsd_language_English ;
  dc:modified "2019-05-15" ;
  pwo:hasFirstStep opredict:Step_Prepare_Input_Data_Files ;
  rdfs:label "Main Protocol v.0.1" ;
  prov:wasAttributedTo opredict:Agent_Remzi ;
```

# EVALUATION

In this section we describe the evaluation of our approach, consisting of two parts: first, we revisit each FAIR principle and explain how the principle is addressed. Second, we applied the traditional ontology validation methodology by answering the competency questions through the execution of SPARQL queries (the concrete queries are available in GitHub repo).

## Addressing the FAIR principles

In order for our workflow to comply with FAIR principles, we checked each FAIR criterion defined in *Wilkinson et al. (2016)*, as identified between parentheses below. First, global and persistent identifiers were assigned to resources defined in the workflow and associated data. Rich metadata for workflow and input and output data were created using HCLS (https://www.w3.org/TR/hcls-dataset/) and FAIR data point specification (F2). In addition, the metadata we generated contains an explicit global and persistent identifier of the data they describe (F3). In order to enable the workflow and the data used to be searched, they were uploaded in a triple-store as a FAIR Data Point (https://graphdb.dumontierlab.com/repositories/openpredict). Data can be queried through SPARQL over HTTP(S) protocol (A1.1). Since the data is not private or protected, we don't require authentication and authorisation mechanism (A1.2). All data and metadata are permanently available at Zenodo (https://doi.org/10.5281/zenodo.3770918) to make the metadata accessible even the data is no longer available (A2). We used RDF and OWL with commonly used controlled vocabularies and ontologies such as Bio2RDF vocabulary, SIO and PROV to model input data and workflows (I1). HCLS dataset specification and FAIR Data Point specification were used to define the metadata and provenance of data (I2). Meaningful links between (meta)data such Bio2RDF links and data and workflow were created (I3). To increase reusability of the workflow, we describe the workflow and its data with community standards such as ML-Schema and P-PLAN (R1). We provide the license (R1.1) and provenance information in the metadata using FAIR data point specification (R1.2), and HCLS specification (R1.3) and PROV.

## Answering competency questions

Besides evaluating whether each FAIR principle was addressed, we also assessed the unified model using the common semantic validation approach, which is based on SPARQL queries used to answer the competency questions. All questions listed in 'The FAIR Workflows Approach' could be answered by running the SPARQL queries over the OpenPREDICT use case. The complete queries and results can be found online (https://github.com/fair-workflows/openpredict). Therefore, the reproduction of this validation can be performed by re-executing the queries on the RDF representation of the OpenPREDICT workflow. Below we explain the result for each competency question.

### CQ1 - Questions about manual steps.
### CQ1.1: which steps are meant to be executed manually and which to be executed computationally?

The SPARQL query we built to answer this question first filters all steps within the first version of OpenPREDICT workflow (*opredict:Plan_Main_Protocol_v01*). The results show each step and its type—manual (*bpmn:ManualTask*) or computational (*bpmn:ScriptTask*)—as well as the respective instructions (*p-plan:Plan*) that describe the steps. In summary, OpenPREDICT v0.1 has 28 manual steps and 14 computational steps (42 in total), while v0.2 has 9 manual steps and 9 computational steps (18 in total). This difference reflects the automatization of most of the manual steps within data preparation (evolving from manual to computational) and the simplification of the computational steps described in fewer Jupyter Notebook cells.

### CQ1.2: For the manual steps, who are the agents responsible to execute them?

To answer this question we filtered the results for only manual steps through the statement:
*values ?stepType bpmn:ManualTask*

The result is a list of all steps and roles related to each one, such as executor, creator, developer, and publisher. For example, Remzi is creator, developer and executor of all instructions, while Ahmed is developer of some computational steps and Joao is the executor of the entire OpenPREDICT workflow. This approach allows for the representation of multiple roles played by different agents within each step.

As in related approaches such as Workflow4ever and Reproduce-me, we use the PROV ontology to address the different types of agents and roles through the *prov:wasAttributedTo* property, and apply the *dc:creator* and *dc:publisher* properties for the direct relation from an instruction to an agent.

### CQ1.3: Which datasets were manually handled and what are their formats?

OpenPREDICT's computational steps use datasets, as explained in 'FAIRified data collection', that required manual pre-processing. The difference between v0.1 and v0.2 is that we automated the manual pre-processing of two datasets in v0.2; MESH Phenotype annotations and protein-protein inter-actions. The main elements of the query reflect the FAIR data point specification with DCAT elements (*dcat:Distribution, dcat:downloadURL* and *dcat:mediaType*), PROV (*prov:Usage* and *prov:qualifiedUsage*) and EDAM classification for data handling steps (*edam:operation_2409*) and data formats (media types).

### CQ1.4: What are the types of manual steps involved, and what are their inputs and outputs?

Similar to the Reproduce-me approach, our ontology leverages on the P-PLAN ontology to address the variables used as input and output of the manual steps, mostly during data preparation in OpenPREDICT v0.1, such as downloading and saving the datasets listed in the results of CQ1.3. For example, the input of *opredict:Step_Save_files_in_triplestore* are variables that indicate the local file of each dataset (serialized as RDF) and the output variable indicating the endpoint to upload all datasets (*opredict:Variable_Triplestore_endpoint_for_input_data*).

When changing the filter from manual steps to computational steps, the pattern followed was to classify the output variables of a step (a Jupyter Notebook cell) according to the data saved in files. For example, in feature generation, the *opredict:Step_Feature_generation_01_Pipeline_Source_Cell11* has an output variable for drug fingerprint similarity, indicating the generation of the file "drugs-fingerprint-sim.csv".

### CQ2 - Questions about instantiation of general workflows by more specific ones.

### CQ2.1: What are the main steps of a general workflow?

OpenPREDICT workflow follows the common machine learning pipeline process of: data preparation, feature generation, model training, model evaluation and presentation of results. The query returns these steps by looking for the first step of the workflow (through *pwo:hasFirstStep*) and following the preceding path in a recursive way, e.g.,

```
?step1 dul:precedes ?step2.
?step2 dul:precedes ?step3.
?step3 dul:precedes ?step4. (until there is no preceding steps)
```

The classification of the step is given by the EDAM specializations of the Operation concept *(operation_0004)*, such as *Data Handling* for data preparation (*edam:operation_2409*). For the sake of simplicity, model training and evaluation were performed within the same step. The main steps are listed below:

```
opredict:Step_Prepare_Input_Data_Files
opredict:Step_Feature_generation_Pipeline_OpenPREDICT_ipynb
opredict:
    Step_Model_preparation_train_and_evaluation_Workflow_OpenPREDCIT_-
    _ML_ipynb
opredict:Step_Format_results_for_presentation
```

### CQ2.2: What are the steps of a specific workflow?

Similar to the previous question, the SPARQL query uses the properties that allow for the ordering of steps execution (*pwo:hasFirstStep* and *dul:precedes*). The pattern *p-plan:Step dul:isDescribedBy p-plan:Plan* allows us to answer this question, by representing how a step is described by an instruction. This pattern resembles the one used by Workflow4ever, which applies the *wfdesc:hasWorkflowDefinition* (*dul:isDescribed*) to link a *wfdesc:Workflow* (*p-plan:Step*) to a *wfdesc:WorkflowDefinition* (*p-plan:Plan*), aiming at representing the instructions (e.g., a Python script) that are natively understood by the *wfdesc:WorkflowEngine* (*prov:SoftwareAgent*). However, different from this approach, we classify the instruction language (*p-plan:Plan dc:language dc:LinguisticSystem*), allowing for the representation of instructions that follow computer language or natural language, which includes pseudo-code—commonly used to specify algorithms before implementing in a particular computer language.

The results show that OpenPREDICT has 78 steps in total, where 60 steps belong to v0.1 and 18 belong to v0.2, each step linked to an instruction. 9 instructions were reused from v0.1 to v0.2 regarding data preparation, thus, v0.2 presents 9 new instructions that are used to automate the data preparation phase. These instructions are written as either English (natural language) or Python 3.5 (computer language), where most of the Python

ones refer to the Jupyter notebook cells for feature generation and model training and evaluation.

### CQ2.3: What higher-level description does a certain workflow step instantiate?

The SPARQL query to answer this question includes the pattern *p-plan:Plan dul:isDescribedBy p-plan:Plan*, which extends the capability described in the previous question, i.e., decoupling steps from instructions, enabling the representation of different abstraction levels of instructions and their relations. This pattern resembles the links between specification artefacts (e.g., conceptual model, activity diagrams and use cases) and implementation artefacts (e.g., software code, deployment procedures and automated tests) in software engineering. Usually, a specification artefact aims at describing the instructions necessary to enable a programmer to create the software code, sometimes automatically generated as in model-driven engineering. For example, a pseudo-code within an activity diagram (*p-plan:Plan*) may describe the behaviour expected (*dul:isDescribed*) for the algorithm behind a service endpoint, which may be implemented as a Python script (*p-plan:Plan*).

OpenPREDICT did not formally follow the specification phase of software engineering since it is a research project, having the code developed from the data scientist interpretation perspective about publications related to PREDICT. In research-oriented data science this type of approach is common. However, we created some examples of the pattern that represent the specification of OpenPREDICT workflow. Therefore, the results of this query include 10 Jupyter Notebook cell instructions (*p-plan:Plan*), representing implementation artefacts, that were specified (p-plan:isDescribedBy) by 3 specification instructions (*p-plan:Plan*). The level of abstraction can be derived from the properties of the instruction. For example, the 10 Jupyter Notebook cell instructions were written (*dc:language*) in Python 3.5 (*schema:ComputerLanguage*), while the 3 specification instructions were written in English (*en* value of *xsd:language*). Furthermore, this approach enables links of *s* (specification artefacts) *x i* (implementation artefacts), where *i¿s*, i.e., a specification artefact usually describes several software code lines (instructions). In OpenPREDICT, the first specification instruction guides the load of input datasets, which is linked to cells 1–5 of the feature generation step, while the second guides the calculation of scores between pairs of drugs and compute similarity feature, which is linked to cells 6–9.

### CQ3 - Questions about versioning of workflows and their executions
### CQ3.1: What are the existing versions of a workflow and what are their provenance?

The collective workflow (the whole) is represented as a *dul:Workflow* and a *p-plan:Plan*. Similar to other approaches (Workflow4ever, Reproduce-me, CWLProv, among others) the query to answer this question makes use of DC properties (e.g., *dc:creator, dc:created, dc:modified*) and PROV (e.g., *prov:wasAttributedTo*) for prospective provenance. It also covers workflow versioning through *dc:hasVersion* and *prov:wasRevisionOf*, where the former is responsible for version of *dul:Workflow* and the latter to link an instruction to another (*p-plan:Plan prov:wasRevisionOf p-plan:Plan* pattern). The retrospective

(executions) provenance is supported by the link from an execution (a *p-plan:Activity*) to the correspondent step (*p-plan:correspondsToStep* property), which is a pattern that resembles most of the aforementioned semantic models. The main difference here is the assumption that any instruction (*p-plan:Plan*) should be versionable, thus, all executions link to a versioned instruction. Differently from Workflow4ever approach, here we do not introduce any elements regarding the specification of the changes (e.g., *roevo:ChangeSpecification*). The results for OpenPREDICT show 2 workflows (v0.1 and v0.2), both created by and attributed to Remzi, where v0.2 links to the prior version (v0.1).

### CQ3.2: Which instructions were removed/changed/added from one version to another?

Three SPARQL queries were written to answer whether the instructions of OpenPREDICT v0.1 were removed or changed or added in v0.2. Each SPARQL uses the identifier of the workflow versions (retrieved in CQ3.1) as an input parameter to perform the comparison from one version to another. For the query for removed instructions, it considers all instructions used in v0.1 that are not used in v0.2 and excludes the instructions that were changed. For the query for changed instructions, it considers the instructions with the *prov:wasRevisionOf* property. For the query for added instructions, the SPARQL query uses the reverse logic from the removed.

Forty-seven instructions were removed from v0.1 to v0.2 due to the refactoring of the code of feature generation, model training and model evaluation, and the elimination of several manual steps in data preparation. Three instructions were changed, reflecting the porting of the FAIRification manual steps to computational steps in data preparation, i.e., download and save human interactome and phenotype annotations. Seven instructions were added in v0.2, where 3 of them represent the new Python scripts for data preparation of the new data sources, other 3 represent the new scripts for feature generation and the remaining for model training.

### CQ3.3: Which steps were automatized from one version to another?

This query is quite similar to the one used for changed instructions (CQ3.2) but it makes explicit that the old version of the instruction used as manual step (*bpmn:ManualTask*) was modified to an instruction used as computational step (*bpmn:ScriptTask*) in the new version. The results confirm the findings from the previous query regarding the 3 instructions that were ported from manual steps to computational steps, namely the data preparation top-level instruction, the FAIRification instructions (download and save human interactome and phenotype annotations). Although our approach covers change management, we face the same challenges regarding the dependency of the developer practices for code versioning. This means that, for example, a developer is free to choose whether to remove files from an old version of the software implementation and add files to the new version, even though these files refer to the same capability or routines. Most of the version controls track the changes when the files (old and new) have the same name and path (i.e., the step identifier), which is a similar approach used here.

### CQ3.4: Which datasets were removed, changed, or added from one version to the next?

This question can be answered by mixing the same query of CQ1.3 (datasets manually used) with the logic used in query CQ3.2, i.e., one SPARQL query to the datasets removed, one for the changed and one for the added. The query results over OpenPREDICT (v0.1 and v0.2) confirm the findings of CQ1.3, where none datasets were removed from the old version to the new, none changed and 2 were added.

### CQ3.5: Which workflow version was used in each execution and what was generated?

This question is answered by using the pattern *p-plan:Activity p-plan:correspondsToStep p-plan:Step*, where the step is part of the dul:Workflow that provides the workflow version. The OpenPREDICT workflow had 14 executions represented with our unified model, exemplifying the execution of some computational steps, i.e., each one a particular Jupyter Notebook cell. Therefore, this approach allows for the representation of multiple executions of each step according to the version of the corresponding instruction. Each execution inherits the properties of *p-plan:Activity*, e.g., the event start and end time points. Furthermore, each execution is associated to the correspondent generated artefacts through the *p-plan:Activity prov:generated opmw:WorkflowExecutionArtifact* pattern, a similar approach of Workflow4ever, which applied the inverse property *prov:wasGeneratedBy*. An artefact generated by an execution can be an evaluation measure of the trained model, such as the model accuracy and recall for that particular execution, i.e., a *mls:ModelEvaluation*. Therefore, OpenPREDICT executions generated the values about the model evaluation measures of accuracy, average precision, F1, precision, recall and ROC AUC. For example, the results show that the model accuracy of v0.1 is 0.83, while v0.2 is 0.85.

This query can be further extended by considering the particular version of each instruction that the executed step implements. In addition, ideally, each output of a Jupyter Notebook cell should be represented as a *opmw:WorkflowExecutionArtifact*, so all generated outputs are stored (similar to ProvBook/Reproduce-me approach). This query can be easily changed to provide aggregations for related analytical questions, such as how often each workflow version was executed.

## DISCUSSION

Before we move on to discuss the encountered reproducibility challenges and other issues, we would like to first highlight the two FAIR perspectives that our approach embodies and demonstrates. Firstly, FAIR applies to the datasets that a scientific workflow consumes and produces, e.g., the protein–protein interactions dataset used by OpenPREDICT, and we need FAIRification approaches to raise existing datsets to this standard. This is the aspect the FAIR principles originally focused on. On top of that, we have here proposed and exemplified a second perspective. This additional perspective regards the workflows' own FAIRification, i.e., the process of aligning them with the FAIR principles, which relies on a semantic modelling approach such as the one described in this paper. Our work therefore

expands the notions of FAIR and FAIRification from the relatively static artifacts of datasets to the dynamic processes of workflows.

## Reproducibility challenges

It was expected that our study would be unable to fully reproduce the accuracy of the method reported in the PREDICT paper due to use of different input datasets. The performance results of this study are lower than originally reported. The PREDICT paper reported an AUC of 0.90 in cross-validation, but using the same gold standard, we could only achieve AUC of 0.83.

We were also able to obtain the drug and disease similarity matrices used in PREDICT from the authors via email request. Given 5 drug–drug similarity measures for 593 drugs and 2 disease–disease similarity measures for 313 diseases, there are resulting 10 features of combined drug-disease similarities. The logistic classifiers were trained with these pre-computed similarity scores and an average AUC of 0.85 was obtained from 10 repetitions in a 10-fold cross-validation scheme. This is still a significant difference from the AUC of 0.90 what the authors reported in PREDICT study. This indicates that there was more likely an error in the design or implementation of evaluation, and not the aggregation of data nor the calculation of drug-drug and disease-disease similarity scores.

While attempting to reproduce the PREDICT study, we faced the following issues, which we have turned into generic recommendations (highlighted in italics).

1. **Insufficient documentation** Essential details concerning the calculation of features were not clearly defined, nor was the software code to perform the calculations provided. *Many details of an experiment, including data sets, processing parameters and applied software and algorithms need to be specified in order to facilitate the replication of the results. A methods section in a scientific article may not be the best place to provide all this information as it is usually limited by size constraints and different organization styles of journals and conference proceedings, leading to a lack of required detail.*

2. **Inaccessible or missing data** Since no data except the gold standard data (drug–disease associations) were given, the features for the PREDICT workflow were reconstructed using the publicly accessible databases DrugBank and KEGG and SIDER. However, we could not check if this resulted in exactly the same datasets. *The original data that were available to the authors could be absent or no longer accessible to others for many reasons. Sufficient data should be published to enable reproducing a study.*

3. **Versioning and change of data** In PREDICT, publicly accessible datasets have been used to construct models and validate hypotheses for prediction of drug indications and drugs were identified by their Drugbank IDs. However, Drugbank IDs are subject to change over time. For example, two drugs (DB00510, DB00313) in the original dataset were merged to the same drug within the current version of the Drugbank. *Results or hypotheses may change as a result of updated input data. In order to reconstruct the original conclusion, it is important to record the version or the date of the data that were used in a study. This is especially important as publicly accessible datasets are increasingly used to construct models and validate hypotheses for prediction of drug indications.*

4. **Execution environment and third-party dependencies** In the PREDICT study, the versions of some software tools, such as the library for semantic similarity calculation, were not specified. *The versions of software libraries, packages and tools used in a workflow should be explicitly mentioned, and an effort must be made to maintain the access to those releases used in the original workflow.*

## Further Issues Encountered

While the execution of the FAIRification process in the OpenPREDICT was straightforward, the semantic modelling of the unified workflow model was challenging. The reuse of existing semantic vocabularies for the representation of our unified model proved to be an extensive task. There are several existing semantic approaches to represent workflows that present reproducibility issues and different conceptualizations, sometimes overlapping in their terminology. Computational workflow languages such as CWL and WDL do not intend to define a semantic representation for workflows that involve both manual and computational steps and enrich the workflows with sufficient metadata to make them FAIR. The prospective part of Workflow4ever implementation (wfdesc (https://raw.github.com/wf4ever/ro/0.1/wfdesc.owl)) has consistency issues such as missing disjointness and licensing elements, besides not conforming to the documentation (e.g., for all elements related to workflow templates). On the other hand, semantic models like DUL, PROV and P-PLAN presented higher quality and common foundations (in DOLCE), while being easier to reuse and extend. Although CWLProv also provides an ontology-based on W4ever semantic models, it is oriented only to retrospective provenance of computational steps, reusing most of the predicates that P-PLAN extends (from PROV). Furthermore, the URI (https://w3id.org/cwl/prov#) does not provide a concrete description of the new predicates (e.g., cwlprov:image) and neither resolves to the RDF model (TBox).

A question that may arise is whether it would be better to create a new ontology from scratch rather than creating a unified model based on the existing ontologies. We believe that high quality semantic models should be reused, taking benefit from the lessons learned. Furthermore, we consider that reusing existing semantic workflow models actually improve semantic interoperability, while creating a new ontology may impede interoperability if it is not accompanied with alignments to the existing semantic models. Therefore, our approach appears to lead to an improved semantic interoperability. Because we reused several semantic models, the competency questions that they target are potentially addressed by our approach. For example, the gap in our approach regarding the representation of change management for versioning can be addressed by reusing some elements from the versioning approach of Workflow4ever, e.g., *roevo:Change*, *roevo:ChangeSpecification* and *roevo:VersionableResource*.

Deciding which type of approach should be used for role representation should be based on the needs for either a fine-grained definition of the role/relator pattern (a reified relationship), such as the *prov:Association* approach, or a simple property, such as *dc:creator*. While the former (1) enriches the definition of the role (an improved representation capability), the latter (2) is less verbose:

```
(1)
?Association prov:agent opredict:Remzi;
prov:hadRole opredict:Creator;
prov:hadPlan ?plan.

(2)
?plan dc:creator 'Remzi' .
```

One of the main challenges is to understand the different terminology used for similar conceptualizations. Although the definitions of terms like plan, process, protocol, procedure, workflow, plan specification and standard operating procedure seem to be the same (or quite overlapping), their meanings become notoriously ambiguous across varied communities. How to grasp these semantic differences is a crucial question that needs further exploration. For example, in the bioinformatics community, the term *Workflow* usually refer to an implemented (computational) piece of software, i.e., a set of programming language instructions, usually developed with a *Workflow Management System* as a *Workflow Application* (*Da Cruz, Campos & Mattoso, 2012*). Meanwhile, in software engineering, the *Workflow* term is usually referred to a detailed business process within the Business Process Modelling (BPM) research. Usually, the BPM languages conform to graphical notations (e.g., BPMN, EPC, ARIS), targeted to human comprehension rather than computational ends (process design/modelling). Additionally, some BPM languages focus on representing, at a lower level of abstraction, the process execution details, e.g., BPEL (process implementation) (*Rosemann & vom Brocke, 2015*). This is a topic extensively covered by Service Oriented Architecture (SOA) initiatives. Several related works target the gap that exists between business process models and workflow specifications and implementations, such as service composition schemes (*Stephan, Thomas & Manfred, 2012*) and formal provenance of process executions and versioning (*Krishna, Poizat & Salan, 2019*). Furthermore, some of these languages provide predicates for forks and conditionals, which were intentionally not included in the unified model since they have a high complexity - it is still a topic under discussion in the CWL community, for example.

In future work we will improve the modelling of manual steps by studying and possibly incorporating predicates from the SMART protocols ontology. We will characterize the abstraction levels of workflows based on multi-level process modelling approaches, such as the widespread adopted APQC's Process Classification Framework (PCF). The PCF provides 5 abstraction levels for process specification, from a high abstraction level to detailed workflow specification: category (level 1), process group (level 2), process (level 3), activity (level 4) and task (level 5). Although this framework aims at providing a methodological approach for business process specification, we should investigate whether the minimal information elements of each level require proper representation in the ontology. We should also consider the challenges of process refinement (''process description in a more fine-grained representation'') (*Ren et al., 2013*). A process refinement mechanism maps and/or derives models from a higher-level specification to a detailed level, equivalent to vertical and exogenous model transformations in model-driven engineering. Typical refinement categories will be investigated, such as activity decomposition principles about event delivery and execution condition transference (*Jiang et al., 2016*; *Muehlen &*

*Rosemann, 2004*). The representation of intentionality of the activities within business processes will also be addressed in future work through goal-oriented semantic process modeling (*Horkoff et al., 2019*), linking goals to activities and roles.

Industry-oriented approaches are also being investigated, such as Extract, Transform and Loading (ETL/ELT) for data warehousing and SQL Server Integration Services, which considers a workflow as a control flow, while a dataflow transforms data from a source to a destination. Furthermore, Product Line Management (PLM) tools should be investigated, especially the ones that cover Laboratory Information Management System (LIMS), which provides important concepts such as Bill-of-Materials (BoM), specifications and their certifications. For example, in PLM a *specification* is a description of raw materials and packaging materials, and semi-finished and finished products. This description may contain product characteristics (e.g., chemical compounds), recipes (e.g., BoM), production methods, quality norms and methods, artwork, documents and others.

Ultimately, initiatives like CWL, the Center for Expanded Data Annotation and Retrieval (CEDAR) (for metadata management) (*Gonalves et al., 2017*) and FAIRsharing.org (for indexing FAIR standards) may be used as building blocks for the envisioned FAIR workbench tool, which can be a reference implementation over a workflow system such as Jupyter Notebook (e.g., a plug-in). Finally, the validation of the reproducibility level of a workflow should consider specific FAIR metrics that take in consideration specific recommendations (e.g., from CWLProv approach) and the practices for higher reproducibility of Jupyter notebooks (*Pimentel et al., 2019*).

## CONCLUSIONS

In this work, we examined how FAIR principles can be applied to scientific workflows. We adopted the FAIR principles to make the PREDICT workflow, a drug repurposing workflow based on machine learning, open, reproducible, and interoperable. From this stems, the main contribution of this paper, the OpenPREDICT case study, which demonstrates how to make a machine learning workflow FAIR and open. To do this, we created a unified model that reuses several semantic models to show how a workflow can be semantically modeled. We published the workflow representation, data and meta-data in a triple store which was used as FAIR data point. In addition, new competency questions have been defined for FAIR workflows and how these questions can be answered through SPARQL queries. Among the main lessons learned, we highlight how the main existing workflow modelling approaches can be reused and enhanced by our unified model. However, reusing these semantic models showed to be a challenging task, once they present reproducibility issues and different conceptualizations, sometimes overlapping in their terminology.

In the future, we envision that the intensive human effort that we had to perform in order to make a workflow FAIR will be taken care of by smart and intuitive workflow tools. As a prototype of such a tool, we are currently developing the FAIR workbench as a general tool that allows users to deal with workflows and protocols in a semantic and FAIR form.

### Funding

This work was supported by the Dutch Research Council (NWO) (No. 628.011.011) and the Netherlands eScience Center (No.NLeSC P 17.0201). The funders had no role in study design, data collection and analysis, decision to publish, or preparation of the manuscript.

### Grant Disclosures

The following grant information was disclosed by the authors:
Dutch Research Council: 628.011.011.
Netherlands eScience Center (No. NLeSC P 17.0201): NLeSC P 17.0201.

### Competing Interests

The authors declare there are no competing interests.

### Author Contributions

- Remzi Celebi and Joao Rebelo Moreira conceived and designed the experiments, performed the experiments, analyzed the data, performed the computation work, prepared figures and/or tables, authored or reviewed drafts of the paper, and approved the final draft.
- Ahmed A. Hassan analyzed the data, authored or reviewed drafts of the paper, and approved the final draft.
- Sandeep Ayyar analyzed the data, performed the computation work, prepared figures and/or tables, and approved the final draft.
- Lars Ridder, Tobias Kuhn and Michel Dumontier conceived and designed the experiments, authored or reviewed drafts of the paper, and approved the final draft.

### Data Availability

Data and code are available at GitHub: https://github.com/fair-workflows/openpredict.

### Supplemental Information

Supplemental information for this article can be found online at http://dx.doi.org/10.7717/peerj-cs.281#supplemental-information.

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
