# Peer review of "Towards FAIR protocols and workflows: the OpenPREDICT use case"

_PeerJ Computer Science, doi:10.7717/peerj-cs.281_

## Round 0.1 · original submission · Major Revisions

The reviewers provided insightful reviews, with great suggestions to improve the overall quality of the article. Therefore, the authors need to extensively revise their article following such comments to make it more readable, to present all the relevant aspect of their work in the appropriate way, and to have it finally accepted in PeerJ CS.

In addition, since this article is about presenting an ontology to model workflows, it would have expected a discussion about the ontologies developed in the past for addressing the same issue (e.g. OPMW, SCUFL2 Core, PWO), with an explanation about why they have not been considered appropriate for being used in this context.

·

Basic reporting

Celebi, Moreira and colleagues report on their experience trying to represent and reproduce the results of a highly cited workflow (PREDICT) and share their learnings when applying the FAIR principles to workflow objects. The authors provide extensive background information on the state of the art and propose a semantic profile for representing essential information and making workflows FAIR. The semantic profile constructed by the authors builds on top of several well established frameworks, thus following best practice by reusing existing model when available and suggested extensions where applicable.
The work is extensive however the major comment on the manuscript is its length. We feel that the key message is diluted in extensive textual narrative. It may therefore be beneficial from changes to slimline some of sections.

clarity and style sometimes gets in the way, for example at line 215: "the data scientists highlighted the need of defining a generic way of computational and manual workflows," -> please check the english or rephrase.

another example: line 630 "Responding competency questions" -> "answering" should be used instead of 'responding'
" Besides evaluating whether each FAIR principle was addressed, we also assessed the profile ontology using the common semantic validation approach, which is based on SPARQL queries used to respond the competency questions.

again s/respond/aswer/ and is it 'profile ontology' or 'ontology profile' ?

In the Abstract, but also in the introduce, could the authors clarify what they mean by 'dynamic workflows' (abstract and at line 190).

Experimental design

The purpose and aim of the study (a case study) are clearly outlined.
Starting from an assessment of the PREDICT workflow, identifying the necessary steps to make a workflow FAIR, from clarifying input data and building a semantic profile to make the workflow 'self-explanatory' and self-describing.

Validity of the findings

The authors present the difficulties faced when attempting to reproduce and execute the PREDICT workflow. They highlight the difficulties of reusing existing semantic workflows, the problems of domain specific semantic and ambiguity in mapping across.

In the introduction, at line 193, the authors indicate having identified 'disturbing issues'. What are those?
If the authors have reached out to the original authors of PREDICT, it would worth highlighting that fact and possibly document the reasons supplied by the authors for not supplying the original data in the first place.

Table 1 should be introduced much earlier (i.e. under the 'FAIRified Data Collection' section). It would be make it clear for the readership that the original data is not available (please confirm). Table 1 should also be expanded to include last acmes dates, release points for PREDICT, openPredict v1 and v2.

Additional comments

A nice manuscript with rich information, interesting topic and insights but somehow undermined by overwhelming background information. I would suggest a leaner write up, more concise with making the path from PREDICT to OpenPredict more clear to the readership, possibly adding a figure highlight the various steps, (e.g. obtention of input data, ETL, building semantic profile and creation of instances.

Below are a number of suggestions and things I have caught.

introduction
* * *
minor:
line 44: "The emergence of public data, fueled by a combination of literature curated knowledge and ‘omics data, that has created exciting opportunities for computational drug repositioning."
drop "that" in the sentence.
suggestion: replace 'emergence' with 'availability'
suggestion: replace 'fueled by', with 'either in the form of literature curated knowledge or omics data....""

line 68: "The main contributions of this paper therefore are (a) general guidelines..."
suggestion: "Therefore, the main contributions of this paper are (a) general guidelines..."



BACKGROUND
Reproducibility of scientific workflows
* * *
(QR1) What are the input and output variables of an experiment (overlaps with QW2 and QW3);

Could the authors clarify the meaning of 'experiment' in such context ? Also what do they mean by 'variables' associated to an experiment?


The FAIR ecosystem:
* * *
line 168:
"For example, to represent the dataset about protein-protein interactions in human interactome (a CSV file), Bioportal provides the “protein-protein interaction” class."

Can the author be more specific here ? my personal query over bioportal retrieves 23 hits, one of which corresponds to a deprecated class.
It would be nice if the authors could explain which ontology ressource

line 176: (minor comment)
there should be a break and a new section created. "The FAIR principles have received significant attention, but little is known about how scientific protocols and workflows can be aligned with these principles."

line 180: what are "big data-type applications" ?

line 187: unclear sentenece: "For example, it guides the re-use well-maintained libraries packages rather than reimplementing;" - please rephrase

line 191: "This is important because the details of a scientific workflow with the different steps and parameters are often imprecisely detailed in scientific publications and therefore cannot be reliably reproduced." This sentence should be backed by evidence or a citation.


line 207:
"The main goal of these short interviews was to align some characteristics required for the FAIR workbench (that rely on our ontology) with the needs of the researchers, especially from the drug discovery and repurposing domains."

The auhors may wish to introduce the notion of FAIR workbench, as it is not defined anywhere. Also, the sentence may benefit from swapping the clauses around, ie:

The main goal of these short interviews was to align the needs of the researchers, especially from the drug discovery and repurposing domains, with some characteristics required for the FAIR workbench , a tool which....and that rely on our ontology".

line 215: "the data scientists highlighted the need of defining a generic way of computational and manual workflows," -> please check the english or rephrase.

line 218: "A unanimous recommendation is to allow the separation of the workflow steps from the workflow hyper-parameters, so that one can run the same workflow many times without changing the workflow itself."

This sentence lacks clarify. The readers should be reminded of what 'hyper-parameters' in the context of machine learning.


The authors should define what a workflow step is and how they differ from a workflow or an instruction.
also, it would be important to give the differentiating properties of a 'manual step' and a 'computational step'

some of the definitions come later in the text but the current organization of the manuscript makes reading hard at times.

Distinguishing between A workflow specification and a workflow execution and local parametrisation (paramter setting for a workflow run)



I have some issues with the following sentence:
Line 282: "It is often the case that a data scientist needs to manually execute computational instructions, a capability that some workflow systems have, such as Jupyter Notebook (execution of instructions cell by cell)."

I am a bit inconfortable calling Jupyter Notebook a workflow system. While it has some features of a workflow execution system, it is according to its description " an open-source web application that allows you to create and share documents that contain live code, equations, visualizations and narrative text. "
Furthermore, Jupyter notebook are very sensitive to cell state and order of execution.
But again, i come back to the issue of providing a clear example of a 'manual step', which can not be subsumed into a "computational one".


line 297: "our approach is leveraged by the best practices on reusing parts of semantic models to build a profile ontology." -> rephrase: what about "our approach follows the best practices, promoting ontology and semantic model reuse to build an ontology profile"



Figure 1:
What kind of representation formalism is used ? it seems ER diagram with UML features but this should be stated as well as the software used to create the figure.


Figure 2:
Poor resolution with blurring and low quality (but this is only true for the figure embedded in the pdf document for review, the originals seem of high enough quality, feel free to disregard the comment)

Line 405: typo: opmw:WorkflowExecutionArtifcat -? opmw:WorkflowExecutionArtifact


Line 435: FAIRified Data Collection

PREDICT was published in 2011 by Gottlieb et al:
https://www.ncbi.nlm.nih.gov/pmc/articles/PMC3159979/

when the authors list the sources of data, it is actually unclear which exact version is actually used. Could the authors clarify that point by providing a release point or a link to the actual source file or endpoint hit for data download?

for instance, HPO provides its annotation files from http://compbio.charite.de/jenkins/job/hpo.annotations.monthly/lastStableBuild/

Could the authors provide a table listing the exact version and build of the source files they have used for that work ?

line 449: "449 The data that are not in linked data format (RDF) were converted to RDF with a FAIRification process Jacobsen et al. (2019)."

Can the authors provide the full list of such data?
The FAIRification process isn't really described for such sources. Jacobsen et al indicate that a semantic model needs to be build. Where are those for such data sources?

the table should be referenced much earlier, making reading much easier and sending a clearer message.

Table 1. All datasets used in OpenPREDICT version v0.2

however this table should be augmented to include the actual dataset versions (and release dates) used in the initial PREDICT paper in order to highlight the differences between PREDICT and OPENPREDICT datasets.


line 666: "OpenPREDICT v0.2 computational steps use more 2 datasets:" -> should be "OpenPREDICT v0.2 computational steps use 2 more datasets:"




Discussion Section:

Major comment:
"Our study was unable to fully reproduce the dataset and accuracy of the method reported in the PREDICT paper."

The main difficulty I ran into when going over the manuscript is understanding which input datasets have been actually used in this reproduce exercice. From that statement, I would assume that Table 1 should collate all the datasets *actually* used in the paper by Gottlieb et al, but that is not the case.

Table 1 therefore holds a list of datasets released after the Gottlieb paper and therefore represent a very different input. There should be not surprise being unable to reproduce the same


Minor comment: English is sometimes an issue hurting the flow, so I would encourage another round of proof reading.
e.g. "During reproducing the PREDICT study, we faced the following challenges" -> "While attempting to reproduce the PREDICT study, we faced the following challenges"

Reproducibility Challenges:


Acknowledgement:
April fool?

Reviewer 2 ·

Basic reporting

Disappointingly, the paper is confusing, muddled and sometimes impenetrable. This makes it difficult read. Material is presented or referred to without context or explanation. The paper is in a confusing order and the background needs to be rewritten. It really about a semantic model, competency questions to build and validate that model.

- The background is a mixture of literature, the model and then some competency questions that are complementary but without actually presenting the questions that they are complementary to, or an example of a workflow. The literature cited in the background is a strange mixture of reproducibility, ontologies and knowledge representation technologies. Why is SHACL relevant here?

- For a paper on FAIR it would be expected to discuss what the exact interpretation of FAIR is by the authors and a clear explanation of what is meant by a workflow.

- The FAIR ecosystem presented as background is the GO-FAIR ecosystem, not the FAIR ecosystem. This is not a problem so long as it’s clearly stated that this is particular ecosystem that the authors are operating in and is used in their methodology, and that other services and approaches are available (it is not THE FAIR ecosystem). It is one take on FAIR.

- The FAIR ecosystem section is (i) a description of the GO-FAIR process in []; (ii) a superficial discussion on FAIR software (as computational workflows are software); and (iii) some discussion about FAIR incorporated into the workflows of scientists, which is different to the issue of describing workflows so that they are themselves FAIR. This is all muddled together.

- The FAIR Workflows Approach section competency questions are presented ok (though here would be the place to compare CQ in a table for example, rather than the background section. The model itself is presented in a dense and confusing way so that it’s difficult to relate to the Profile and you have yet to state a concrete example of the workflow. Muddling up references to other work just makes it even more confusing. This whole section needs to be grounded in a concrete example – like the PREDICT workflow. The whole section on SHACL (line 357) is obscure – what is the intended point of this?

- The PREDICT workflow needs to be presented earlier and used throughout

- The evaluation on “responding competency questions” is cluttered with references to CQ in other approaches.

- The discussion confuses the representation of PREDICT with the completeness of the PREDICT model. Some of the discussion doesn’t seem to relate to the modelling exercise but more generally (points 3 and 4)

- The section “foundations of FAIR workflow semantic profile” should be moved to the presentation of the profile.

- The role of CWL, CEDAR and FAIRsharing are thrown into the discussion (no citations) without context. These should be incorporated properly.

- Many English issues need to be fixed (for example, there is no such word as “identificator”).
- All Acronyms should be explained. All cited systems should be referenced.

It is not clear who is the intended readership. If it is computer scientists then there needs to be more explanation of the experimental steps undertaken by life scientists and the tools they use. If it is a life scientist then the semantic model for FAIR workflows needs much more careful and simpler presentation. If it is an ontologist or semantic modeller then the discussion about the model is interesting but the narrative mixes up the model with the sources of the model makes it more laboured than need be and obfuscates the main messages. One assumes that this is the intended audience as otherwise terms like “endurant” would not be used, and the separation of the profile from the model would not have been presented. Figure 1 is not clear unless the reader is an expert ontologist

The literature is referenced but confusingly presented and terms are not always defined –for example, what is the difference between a parameter, a meta-parameter and a hyper-parameter? Given that the paper is about workflows it is necessary to explain what is meant by a workflow and give an example of one (this doesn't happen until the PREDICT case study, and this is too general). The paper mentions workflow languages and systems, but it appears that the actual example is a Jupyter Notebook – this is reveal incidentally. (By the way, Jupyter Notebooks do not enforce rigorous stepwise executions – one can bounce between cells, so the step order is only recorded by log trails and that requires a system like ProvBook, which is not a native JN subsystem).

Experimental design

The research question is partially defined. The abstract implies that the paper will be about FAIR principles applied to workflows. The paper is about defining a semantic model for steps of instructions in experimental practice in scientific investigations using an example for exposition and validation. This is a necessary step towards FAIR workflows but the claim that the paper presents general FAIR guidelines seems a little exaggerated.

Appendix 1 gives details of the interviews. The interview methodology for the competency questions should be sketched in the paper. Although the number of researchers is a low number (6) this might be sufficient if the selection criteria and range of skills of the interviewees was clearer. None appear to have used computational workflow platforms. Many appear to be from the authors laboratory (a high incidence of SPARQL experience is unusual). It is unclear to the extent they are wet lab (using lab protocols/SOPs) or dry lab (using computational frameworks). This leads to some suspicions of a rather skewed interview cohort.

The interview results were interesting and although the competency questions make sense it was not entirely clear how the competency questions came out of the interviews. Were the standard practices of thematic analysis - coding semi structured interview transcripts - undertaken? How selective were the questions? For example, interviewees highlighted the need to describe parameters “what are the parameters for workflow X” but this did not appear as a competency question. Other CQs are not adequately explained – for example, CQ2.4 (line 236) which is not presented in the subsection on “responding competency questions” (line 630) so this reviewer could not work what it means by examining an example.

The methodology of “FAIRified Data Collection” (line 435) starts well enough and in general can be followed but many major steps are glossed over and I am sure it is not reproducible from this paper. We have to take on word that the section on “addressing the FAIR principles” (line 613) is achieved. The FAIRifier approach is not well enough described and the RDF snippet using the profile ontology is trivial.

Line 436 – why is it necessary to achieve FAIR data by generating Linked Data? Is this “FAIRification” or RDF publishing? The step “define a semantic model for the data” (line 459) seems like a major activity akin to semantic harmonisation and integration and a recognised global challenge. How hard was this? Where did you “search” for “interactor and source concepts”?

Validity of the findings

The model is interesting and the CQs are a worthwhile contribution, though is they address “previously unaddressed reproducibility requirements” is unclear -which previously unaddressed requirements? The evaluation rests in the capability of the model to answer these CQ.

The abstract implies that the paper will be about FAIR principles applied to workflows – business processes, computational workflows / scripts, and manual protocols. In fact there is little about “FAIR” as such – how workflows and protocols may be found, accessed, interoperated and reused. Computational workflows are also software artefacts that challenge the FAIR data principles especially with regard to interoperability and reuse – see (Anna-Lena Lamprecht, et al Towards FAIR principles for research software, Data Science, vol. Pre-press, pp. 1-23, 2019 DOI: 10.3233/DS-190026) for discussions on FAIR principles applied to software and a number of papers in the recent special issue on Data Intelligence (http://www.data-intelligence.org/p/67/), for a wider discussion on the applicability of FAIR to workflows, protocols and e-infrastructure.

The paper is fundamentally about defining a semantic model for steps of instructions in experimental practice in scientific investigations using an example for exposition and validation. The contribution for “general guidelines to make scientific workflows open and FAIR” is not really substantiated.

- Line 924 “in this work we examined how the FAIR Principles can be applied to scientific workflows”. This is not really substantiated.
- Line 924 “we adapted the FAIR principles to make the PREDICT workflow [ ] open and reproducible”. This is not fully substantiated. Reproducibility of software and method is more that describing a process in RDF. Did you mean “adopt” rather than “adapt”?

CWL (and WDL) are recognised workflow description languages (adopted by GA4GH and others). The authors state that CWL is the “de facto standard for syntactic interoperability of workflow management systems” line 93), and that CWL has a Prov profile to define how to record provenance of a workflow run, (https://github.com/common-workflow-language/cwlprov, https://doi.org/10.1093/gigascience/giz095 it would be useful to compare the profile on page 7 with CWL (there are some references to CWLProv).

As the model covers Lab Protocols and SOPs, it is surprising that models such as Smart protocols (https://bioportal.bioontology.org/ontologies/SP), Autoprotocol (http://autoprotocol.org) and Procotols.io are not discussed.

It is not clear the range of expressivity of the workflow that can be expressed. How are forks and conditionals addressed? The implication is that these are simple pipelines.
The profile (Figure 1) seems entirely generic except for the Machine Learning part. Why did you make the decision to explicitly include ML and not, say, a simulation sweep or an ODE model construction step?

Additional comments

In this paper there is some very good and interesting work.
The work to apply FAIR to workflows is important. The separation of “instruction” from the “step” is particularly insightful and the approach of interviews and CQ is very good. the Interviews were insightful

However, the paper needs to be substantially rewritten to be publishable – everything is thrown together and the messages are lost or confused or both. The current presentation does not do your work the credit it deserves.

- Who is the audience you intend to address? And is the background and terminology suitable for that audience?
- Restrict your claims to the presentation and evaluation of a model for workflows
- Be clear and clean about what you mean by workflows, by FAIR and the model itself.
- Suggested reorganisation:
o background presents literature on FAIR data and software and explain your interpretation of the FAIR principles;
o present the workflow landscape: manual protocols, business processes, computational workflows (interactive and non-interactive) and their operating environments (notebooks, scripts, BPMs, WfMS etc).
o then introduce the PREDICT workflow as a running example. Present the TWO OpenPREDICT versions diagrammatically so we can understand them when we work through the representation later.
o present the competency questions, with a table that shows the relationship to other questions and highlights the novelties, and how these map to the FAIR principles
o the semantic model is then presented without obscuring the model with the profile (or clean this up – the profile is one place and the model in another) with concrete examples using the PREDICT workflow. The section “foundations of FAIR workflow semantic profile” should be moved to here and repetition removed.
o then have a section on FAIRifying the data collection which, as you are following just one methodology, you can introduce here and work through concretely. Currently we have to take on word that the section on “addressing the FAIR principles” (line 613) is achieved.
o Then present the workflow in the model relating the description to the visual steps or annotating the description. Be clear that this is a Jupyter Notebook. Maybe even show the notebook
o In the evaluation do not clutter the “responding competency questions” with references to CQ in other approaches. Include all the CQs (2.4 is missing)
o Sort the discussion into (i) the ability of the model to represent the expressivity of the workflows faithfully (ii) the completeness of the representation and (iii) other topics regarding workflow FAIRness and reproducibility not addressed by the model or by a static semantic representation of processes.

Other comments
Line 144 – are the FAIR principles are not a minimal set of requirements? They seem to be more an aspirational set of principles and it is unlikely that all principles will be implemented for any dataset and therefore unlikely to be minimal. See Mons et al Information Services & Use. 37. 1-8. 10.3233/ISU-170824)

The FAIR ecosystem (Line 143) is the GO-FAIR ecosystem, rather than the FAIR ecosystem. GO-FAIR metrics is one of several initiatives developing indicators for measuring principles, notably the RDA FAIR Data Maturity WG. There are numerous initiatives developing FAIR services. This GO-FAIR ecosystem is oriented to publishing RDF, which is an expensive step – actually the FAIR principles do not pertain to any technology and argue for machine processability and standards – JSON and Schema.org would qualify as well as anything. Other FAIR pipelines annotate with 'standard' vocabularies rather than turn everything into RDF

The Hodson interim report (line 975) has been superseded by the final report (DOI: 10.2777/1524).

Line 488 references Figure 4.1 - there are only figures 1 and 2.

Line 940, Acknowledgements. Perhaps you should acknowledge Douglas Adams?

---

## Round 0.2 · accepted · Accept

The authors did a huge work in the revision they submitted for review and they addressed almost all the comments and observations provided by the reviewers.

There are still a few points of attention that should be addressed – see the comments of the reviewer attached. However, I think they can be considered during the preparation of the camera-ready version of the article. To summarise them up, they are:

- the connection from DOLCE to the description of workflow systems needs a bit more of argumentation;

- CWL and WDL are some of the most supported semantic models for workflows and should be appropriately mentioned and discussed;

- properly highlight and discuss the two FAIR perspectives about workflow, i.e. their role in data FAIRification and their own FAIRification;

- address some overclaims (e.g. the abstract);

- carefully check the English of the whole article.

Again, congratulation for your acceptance!

Reviewer 2 ·

Basic reporting

The paper is much clearer than the first draft. Its audience is now clearly that of a semantic modeller. The presentation is tighter and has been reorganised to be clearer and logical. the CQs are much clearer as is the purpose of the paper.

The model is much easier to understand.

A few niggles.

The definition of workflow in the background could still benefit from some attention. I am not convinced the DOLCE definition is the most useful. There is a big jump from this to the description of workflow systems. The main thing about a workflow is the separation of the workflow specification from its execution, be that manual (so called Human in the Loop) or automated.

The paper focuses on provenance of a workflow when reviewing past models but not so much the description of the workflow before execution (so called "prospective provenance"). The discussion of CWL still seems cursory, focussing on CWLProv. CWL and WDL (still not mentioned) are the most widely supported common semantic model for workflows in open science and yet rather dismissed (or ignored). Which is a pity.

In the meantime the BioCompute IEEE standard 2791-2020, has been published (https://opensource.ieee.org/2791-object/ieee-2791-schema/).

Section 2.3 (Applying FAIR to Workflows) still seems to focus more on the FAIRificaton of data to be consumed (or produced?) by the workflow than on FAIR workflows as objects in themselves (which is the main focus of the model presented). [1] highlighted these two sides of FAIR for workflows - the role they play in data fairification and their FAIRification as artefacts in their own right.

The paper still has some typos and odd spellings that should be addressed.

[1] C. Goble, S. Cohen-Boulakia, S. Soiland-Reyes, D. Garijo, Y. Gil, M.R. Crusoe, K. Peters & D. Schober. FAIR computational workflows. Data Intelligence 2(2020), 108–121. doi: 10.1162/dint_a_0003

Experimental design

The rewriting of the case study, making it clearer which datasets were used) makes it easier to judge the design. The focus on the CQs and the use of the Case Study for evaluating the model is much more convincingly presented.

Validity of the findings

The paper is far more convincing about its results than previously as it now recognises that this is a step towards FAIR workflows and has toned down its claims.

If the paper is thought of as an exercise in a model and the ability of the case study to answer to competency questions then the findings are valid.

There is still the odd niggle - the abstract claims that "In this paper we describe an inclusive and overarching approach to apply the FAIR principles to workflows and protocols and demonstrate its benefits". This is still overclaiming. The section 5.1 is a little cherry picking in the choice of FAIR principles to comply with (e.g. F4. (Meta)data are registered or indexed in a searchable resource, or maybe that is the FAIR datapoint).

The CQ are a little bit cherry picked too. CQ2.4 still doesn't get a response in section 5 and was not answered in the rebuttal.

Additional comments

I thank the authors for the detailed rebuttal and for responding to the recommendations and comments. The paper is much better and the model and competency questions are a valuable contribution to the field.